# Rollout of the 2022/2023 Seasonal Influenza Vaccination and Correlates of the Use of Enhanced Vaccines among Italian Adults

**DOI:** 10.3390/vaccines11121748

**Published:** 2023-11-23

**Authors:** Luca Pestarino, Alexander Domnich, Andrea Orsi, Federico Bianchi, Elisa Cannavino, Pier Claudio Brasesco, Gianluca Russo, Simone Valbonesi, Giacomo Vallini, Matilde Ogliastro, Giancarlo Icardi

**Affiliations:** 1Private General Practice, 16122 Genoa, Italy; dottpestarino@gmail.com (L.P.); drbianchifederico@gmail.com (F.B.); mmg.cannavino@icloud.com (E.C.); brasesco@fastwebnet.it (P.C.B.); gianlucarussommg@gmail.com (G.R.); valbonesi.simone@gmail.com (S.V.); 2Medicoop Liguria, 16122 Genoa, Italy; giacomo.vallini@webmit.eu; 3Hygiene Unit, San Martino Policlinico Hospital-IRCCS for Oncology and Neurosciences, 16132 Genoa, Italy; andrea.orsi@unige.it (A.O.); icardi@unige.it (G.I.); 4Department of Health Sciences (DISSAL), University of Genoa, 16132 Genoa, Italy; matilde.ogliastro@hsanmartino.it; 5Interuniversity Research Center on Influenza and Other Transmissible Infections (CIRI-IT), 16132 Genoa, Italy

**Keywords:** influenza, vaccination, influenza vaccine, adjuvanted influenza vaccine, high-dose influenza vaccine, vaccination uptake, older adults, Italy

## Abstract

In Italy, several types of seasonal influenza vaccines (SIVs) are available for older adults, but for the 2022/2023 season there were no guidelines on their specific use. This cross-sectional study assessed the frequency and determinants of the use of enhanced (adjuvanted and high-dose) SIVs in Italian older adults, as compared to standard-dose non-adjuvanted formulations. Of 1702 vaccines administered to a representative outpatient sample of adults aged ≥ 60 years and residing in Genoa, 69.5% were enhanced SIVs. Older age (adjusted odds ratio (aOR) for each 1-year increase 1.10; *p* < 0.001), and the presence of cardiovascular disease (aOR 1.40; *p* = 0.011) and diabetes (aOR 1.62; *p* = 0.005) were associated with the use of enhanced vaccines. Compared with the adjuvanted SIV, subjects immunized with the high-dose vaccine were older (aOR for each 1-year increase 1.05; *p* < 0.001) and had higher prevalence of respiratory diseases (aOR 1.85; *p* = 0.052). Moreover, usage of the enhanced SIVs was driven by the period of immunization campaign, place of vaccination and physician. Despite their superior immunogenicity and effectiveness, the adoption of enhanced SIVs in Italy is suboptimal, and should be increased. Enhanced formulations are mostly used in the oldest, and in subjects with some co-morbidities.

## 1. Introduction

Although vaccination is the most effective public health means to reduce the tremendous socioeconomic burden of influenza [1,2], which goes far beyond respiratory illness [3], vaccination coverage is still low in most industrialized and developing countries [4,5]. Italy is not an exception: the official estimates suggest [6] that only 20.2% and 56.7% of the general population and older adults, respectively, were vaccinated in the last season. These numbers are well below the minimum achievable and optimal goals of ≥75% and ≥95%, respectively, in all principal at-risk population strata, including older adults, subjects of any age with co-morbidities, young children, and some others [7,8]. The underlying reasons for this low uptake are multiple and, among other factors, include lack of confidence in the vaccine, low perceived need for vaccination, low perceived risk of influenza, missing recommendations from health providers and inadequate social marketing campaigns [9,10]. From the point of view of subject characteristics, the age and presence of co-morbidities have the largest effect on influenza vaccination uptake in Italian adults [11].

As of the 2022/2023 season, several types of seasonal influenza vaccines (SIVs) were available in both Europe [12,13] and the United States (US) [14]. The licensed SIVs are typically quadrivalent, but differ in terms of inactivation (inactivated vs. live attenuated) and purification (split vs. subunit) patterns, production platform (egg-based vs. cell-based), use of adjuvants (adjuvanted vs. non-adjuvanted), antigen amount (standard-dose vs. high-dose), mode of administration (intramuscular vs. intranasal) and age indication [12,13,14]. For the 2022/2023 season in Italy, the following SIVs were available: split and subunit egg-based standard-dose (QIVe; age indication: ≥6 months), cell-based (QIVc; age indication: ≥2 years), live attenuated spray (QLAIV; age indication: 2 < 18 years), adjuvanted with MF59 (aQIV; age indication: ≥65 years) and high-dose (hdQIV; age indication: ≥60 years) quadrivalent SIVs [12]. For that season and according to the national guidelines [12], each available and age-appropriate SIV could be used with no preferential recommendation for a specific SIV. This was in contrast, for example, with the US guidelines [14], in which both aQIV and hdQIV (together with a recombinant vaccine that is licensed [12], but not commercialized, in Italy) are preferentially recommended for older adults. Both aQIV and hdQIV, also known as “enhanced” quadrivalent vaccines (eQIVs) [15], are indeed more immunogenic [16,17,18] and effective [19,20,21,22] than standard-dose quadrivalent vaccines (sdQIVs). For instance, in the pooled analysis of 11 studies, the high-dose vaccine was 14.3% (95% CI: 4.2–23.3%) more effective than standard-dose formulations in preventing probable or laboratory-confirmed influenza-like illness [20]. Analogously, a meta-analysis on the relative effectiveness of the adjuvanted SIV has estimated [21] that the latter was 13.9% (95% CI: 4.2–23.5%) more effective than QIVe in preventing influenza-related medical encounters. Despite their higher price, both eQIVs have been shown to have good value-for-money (i.e., cost-effective or cost-saving compared with sdQIVs) in several jurisdictions, including Italy [23,24]. Conversely, the evidence comparing aQIV and hdQIV is limited to observational studies that yielded inconclusive results [25].

A large amount of research has focused on determinants of SIV uptake in different target groups, including older adults (reviewed in [26,27,28,29]). These correlates may be ascribed as enablers and barriers, which are in turn related to socio-structural, intermediary, healthcare provider, and other domains. By contrast, in the context of a differentiated SIV market, little is known about what factors drive the selection of a specific SIV type. This is particularly relevant for older adults, a target group for which annual vaccination is strongly recommended [1,12,14], and to whom almost all (except for QLAIV) vaccine types may be administered [12,13,14]. On the other side, taking into account the appropriateness of the use of SIVs [30,31], the US recommendations [14] and available comparative research evidence [16,17,18,19,20,21,22], eQIVs seem to be a more suitable choice for older adults. This study therefore aimed to quantify the usage and associated correlates of eQIVs in older adults.

## 2. Materials and Methods

### 2.1. Study Design and Population

This cross-sectional study was part of a larger project on the evaluation of the efficiency and effectiveness of the 2022/2023 season influenza immunization campaign [32,33]. Briefly, the study was conducted in the Metropolitan City of Genoa (Liguria, Northwest Italy) during the 2022/2023 influenza season. Among other Italian (as well as European) cities, Genoa had the highest incidence of the elderly, with about one-third of the total population represented by older adults [34]. The Liguria region is among few Italian regions in which all SIV types are used to a significant extent. Indeed, according to the 2022 regional allotments, 36% of all vaccine doses planned to be purchased were eQIVs. Considering both the current age indication of eQIVs (≥60 and ≥65 years for hdQIV and aQIV, respectively) and free-of-charge recommendation of SIV (any SIV type) for all individuals aged ≥ 60 years independently of other risk conditions [12], the eligible population included older adults aged ≥ 60 years vaccinated with either available SIV. Otherwise, no exclusion criteria were set.

To scrutinize factors associated with the use of eQIVs, we thought to include at least 1000 SIV exposures a priori. With approximately 20 independent variables used to populate a multivariable logistic regression model, the analysis would likely have sufficient statistical power [35,36]. By assuming that an average general practitioner (GP) follows 1200 adults of which 40% are those aged ≥ 60 years, and an SIV coverage of 55% [6] in the latter population, data from at least four GPs were needed to achieve the required sample size. To participate in the study, GPs had to use all available SIV types.

Vaccination status, SIV brand and vaccination date were available in GP’s electronic health records. To capture vaccinations performed outside a GP’s office, data were also cross-checked with a local health unit electronic vaccination registry [32].

The study protocol was approved by the local Ethics Committee (# 166/2023, ID 13094).

### 2.2. Study Variables

For the main analysis, the independent variable of interest was administration of an eQIV (either aQIV or hdQIV) in subjects aged ≥ 60 years. We also conducted a secondary analysis by comparing hdQIV and aQIV users. Considering the different age indication of the latter vaccines (≥60 and ≥65 years, respectively) [12], this analysis was limited to individuals aged ≥ 65 years.

The following were independent variables of interest: (i) sex; (ii) age; (iii) presence of cardiovascular, respiratory, kidney, liver and rheumatic diseases, diabetes mellitus, cancer and other immunosuppressive conditions, obesity, anemia and dementia; (iv) place of vaccine receipt (GP office, community pharmacy or other places). Indeed, although most vaccinations are performed by GPs, some doses are also administered at pharmacies, territory public health units, etc. [32]. Moreover, to account for eventual differences in the temporal distribution of single vaccine types and the propensity of single GPs to vaccinate with a given vaccine formulation, all models were also adjusted for dummy variables of the week of the immunization campaign and the GP.

### 2.3. Statistical Analysis

For descriptive purposes, categorical variables have been reported as percentages with Clopper–Pearson’s exact 95% confidence intervals (CIs), while approximately normally distributed continuous variables were summarized as means with standard deviations (SDs). Categorical variables were compared by means of the Fisher’s exact or chi square tests, while continuous variables were compared by applying the independent *t* test. Multivariable logistic regression analysis was finally computed to establish adjusted odds ratios (aORs) on the association between the use of eQIVs and independent variables of interest (see above). Model selection was performed by minimizing the Akaike’s information criterion (AIC). Nagelkerke’s pseudo-*R*^2^ was computed to illustrate the explained variance of the models.

Data were analyzed in R v. 4.1.0. (R Foundation for Statistical Computing; Vienna, Austria).

## 3. Results

Of the initial 7136 records from five GPs, a total of 2185 unique SIV administrations were identified. Of these latter, 1702 records corresponded to individuals aged ≥ 60 years and were included in the study. The estimated 2022/2023 SIV uptake in older adults was 54.6% (95% CI: 52.8–56.3%). Older age and presence of several comorbidities (cardiovascular and liver diseases, diabetes, anemia, cancer and other immunosuppressive conditions) were independently associated with the uptake of any SIV type (Appendix A).

The main characteristics of the vaccinated adults aged ≥ 60 years are reported in Table 1. Briefly, their mean age was 75.5 (SD 8.9) years and females slightly prevailed (58.9%). About three-quarters (76.3%) of older adults had at least one underlying health condition, of which cardiovascular diseases (64.2%) and diabetes (19.6%) were the most prevalent.

Within single vaccine types, aQIV (761/1702; 44.7%; 95% CI: 42.3–47.1%) and hdQIV (422/1702; 24.8%; 95% CI: 22.8–26.9%) were administered more frequently than sdQIVs (519/1702; 30.5%; 95% CI: 28.3–32.7%). As for immunization campaign rollout, most vaccine doses were administered by mid-November 2022, with a peak of administrations registered during week 43 (24–30 Oct 2022) (Figure 1). The GP practice was the most prevalent (1492/1702; 87.7%; 95% CI: 86.0–89.2%) physical place of vaccination, while SIV administration at pharmacies (88/1702; 5.2%; 95% CI: 4.2–6.3%) and other places (122/1702; 7.2%; 95% CI: 6.0–8.5%) was less common.

Table 2 compares subjects immunized with eQIVs and sdQIVs. eQIV users were on average six years older (77.2 vs. 71.5 years; *p* < 0.001), had a higher prevalence of co-morbidities (81.8% vs. 63.6%; *p* < 0.001) and were more frequently vaccinated by GPs (92.1% vs. 77.6%; *p* < 0.001). Compared with aQIV, hdQIV users had a higher prevalence of chronic respiratory conditions, and no hdQIV doses were administered at pharmacies (Table 2).

According to the multivariable analysis (Table 3), each 1-year increase in age was associated (*p* < 0.001) with a 10% increase in the odds of receiving eQIVs. Subjects with cardiovascular disease (aOR = 1.40; *p* = 0.011) and diabetes (aOR = 1.62; *p* = 0.005) received eQIVs more frequently. By contrast, eQIV users showed decreased odds of receiving vaccination at pharmacies (aOR = 0.40; *p* < 0.001) and other places (aOR = 0.16; *p* < 0.001). Notably, both the factors of the week of vaccination and the GP were significant and alone explained 5.6% and 14.2% of variance, respectively. Each consecutive week of the vaccination campaign was associated (*p* < 0.001) with decreased odds (aOR 0.87; 95% CI: 0.82–0.92) of eQIVs administration. No significant interactions between age and sex, or between age and presence of diabetes or cardiovascular diseases, were found.

We finally compared hdQIV and aQIV users. Considering the current age indication of aQIV, this analysis was restricted to subjects aged ≥ 65 years. Subjects immunized with hdQIV (*N* = 360) were on an average three years older (80.0 vs. 77.2 years) than those vaccinated with aQIV (*N* = 754), with an aOR for each 1-year increase in age of 1.05 (95% CI: 1.04–1.07; *p* < 0.001). hdQIV users also showed a twice as high prevalence of chronic respiratory conditions (7.2% vs. 3.7%), showing an aOR of 1.85 (95% CI: 0.99–3.44; *p* = 0.052). Other variables were not associated with the use of either vaccine.

## 4. Discussion

This study was among the first to investigate determinants of the use of the so-called “enhanced” influenza vaccines, and may be of interest for policymakers, since it was conducted in a region in which all SIV types were available to a significant extent, and in a season in which no national preferential recommendation [12] for either specific SIV type was in place. The choice of a more appropriate SIV type for older adults is still controversial, and recommendations issued by different countries differ [14,25,37,38,39]. As we mentioned earlier, both hdQIV and aQIV are preferentially recommended in the US [14]. Similar recommendations exist in the United Kingdom (UK) [38]. Conversely, in Canada and Germany, only hdQIV is preferentially recommended to the elderly, and this recommendation is primarily based on the available experimental evidence of the superior efficacy of hdQIV over QIVe [25,37]. Still other countries have no explicit guidelines on the choice between several SIV types [39]. These latter countries could point to a higher SIV uptake at a lower overall SIV campaign cost [40]. This strategy, however, does not fit with the appropriate use of the available vaccines [30,31]. Here, we showed that the uptake of eQIVs was suboptimal, as only about 70% received aQIV or hdQIV in the 2022/2023 season. Hopefully, this proportion will increase in the 2023/2024 season. Indeed, differently from the previous season [12], for the 2023/2024 season, the Italian Ministry of Health suggested [41] that among available and age-appropriate vaccine formulations, both aQIV and hdQIV are the only two recommended options for older adults aged ≥ 65 years. These recommendations are aligned with those issued by the US [14] and UK [38] authorities. On the other hand, this preferential recommendation will not necessarily lead to the universal adoption of eQIVs by local authorities. As proof, some years ago (e.g., for the 2018/2019 season), a trivalent version of aQIV (aTIV) was preferentially recommended for Italian adults aged ≥ 75 years. This recommendation, however, was not adopted by several regions [31,40]. Indeed, the official recommendations are not mandatory (e.g., for reimbursement purposes), and vaccinating GPs are still free to select any available SIV type.

GP advice is a well-known positive predictor for SIV uptake [26]. Our findings suggest that eQIVs were more frequently administered by GPs, while sdQIVs showed a higher probability of being administered at pharmacies. This result underlines the crucial role of GPs in providing patient-centered vaccination counseling and, arguably, assuring a more appropriate vaccine choice. As proof, Boccalini et al. [31] reported that about 90% of Italian GPs believe that the available SIVs are different, and that some of them are more appropriate for particular population groups, that some SIVs are more immunogenic and may provide better protection to older adults, and that there is a need to establish explicit guidelines on the appropriateness of the use of the available SIVs in order to reduce the burden of influenza and optimize GPs’ work. On the other hand, that survey [31] has also highlighted that in the last season, only 50.3% of GPs had a sufficient number of all available SIV types. In line with the suboptimal use of eQIVs, local public health authorities should ensure the ready availability of appropriate vaccines by the beginning of the SIV campaign. Importantly, 76.6% of GPs believed that the availability of explicit guidelines on which SIV type to administer to a given population group may reduce the off-label use of single vaccines [31]. In our study, for example, about 1% of all aQIV doses were administered off-label.

eQIVs were more frequently administered to subjects of older age and affected by diabetes and cardiovascular conditions. Mannino et al. [42] reported that many Italian GPs recommended aTIV to more frail, older adults, such as the oldest and those with multiple co-morbidities. Similarly, a survey by Stuurman et al. [39] documented that apart from vaccine availability issues, older age and the presence of multiple health conditions are the key factors that influence the choice of a particular SIV type by GPs in Italy and some other European countries. Specifically, only 69.8% of Italian GPs had more than one SIV type available for older adults, and some of them pointed out insufficient numbers of aTIV doses provided by local health authorities. GPs also declared that compared with older adults aged ≥ 75 years, aTIV was less frequently administered to adults aged 65–74 years (85.5% vs. 50.7%) [39]. In this regard, it should be stressed that eQIVs provide consistently high protection across different age and co-morbidity subgroups. For instance, during three consecutive (from 2017/2018 to 2019/2020) influenza seasons, aTIV was more effective (*p* < 0.05) than sdQIVs in the prevention of influenza-related medical encounters in adults aged 60–74 (by 20.8–24.8%), 75–84 (by 23.5–31.6%) and ≥85 (by 12.8–28.7%) years [43]. Lu et al. [44] analogously reported that the advantage of a trivalent high-dose SIV over standard-dose formulations increased with age, being higher in subjects aged ≥ 85 years. Finally, aTIV was generally more effective than sdQIVs in both older adults with cardiovascular disease and diabetes [45].

Our findings thus have implications for future vaccine effectiveness research, such as test-negative case–control studies. Indeed, both the temporary distribution and propensity of single GPs to use single SIV types varied. These potential confounders should therefore be considered in both absolute and relative vaccine effectiveness studies. Similarly, as the propensity to administer eQIVs varies by healthcare provider, it is essential to ensure the availability of up-to-date continuous medical education (CME) programs that target not only GPs, but also other healthcare providers (e.g., specialist physicians, pharmacists and nurses). In the above-mentioned survey by Boccalini et al. [31], it emerged that, in the previous year, 71% of physicians had not taken part in any influenza-related CME activity. Vezzosi et al. [46] analogously noted that only 38.9% of GPs in Northern Italy were aware of the minimal recommended SIV coverage rate in older adults of 75%. In this regard, de Lusignan et al. [47] conducted a study on the knowledge, attitudes, practices and beliefs about seasonal influenza and SIV among 159 UK GPs and 189 patients aged ≥ 65 years. GPs were surveyed before and after a CME module covering both disease and vaccination with aTIV, while patients were surveyed before and after a routine visit with their GP who participated in the CME activity. Following CME completion, there was a significant increase in GP’s confidence in their ability to address patients’ concerns about seasonal influenza and SIV. Significantly more GPs would have recommended the aTIV enhanced vaccine. Analogously, patients reported an improved confidence in the effectiveness and safety of aTIV after meeting their GP. More importantly, the overall SIV coverage in that study was 82.2%, which is higher than the English average uptake of 72.0% reported in that season. CME activities will have even higher importance in the upcoming years, as several new-generation SIVs are in late clinical development [48,49].

The involvement of pharmacies in the roll out of the SIV campaign is an opportunity to increase vaccination coverage. In our study, about 5% of all vaccine doses were administered at pharmacies, and this percentage will likely increase in the upcoming seasons. Indeed, community pharmacies are easily accessible for all individuals who can enter a pharmacy without a previous appointment and receive professional advice (including SIV counseling) immediately. In a large representative survey of Italian adults [50], community pharmacists ranked as the third (after GPs and healthcare institutions) most credible sources of information regarding SIV. However, SIV administration at a pharmacy was associated with lower odds of receiving an eQIV. Considering their increasing role in vaccination delivery, CME courses on SIV and the appropriate vaccine use should target pharmacists as well. In this regard, one Italian survey documented an insufficient knowledge and lack of interest in SIV among pharmacists [51].

We identified three main study limitations. First, similarly to other registry-based studies, our analysis relies on the quality of data entry. Although the electronic immunization registry used has been previously validated [32,33] and the remuneration of GPs for vaccine doses administered is performed on the basis of their registration in the electronic immunization registry, we cannot preclude that some small number of vaccinations were not registered. We tried to reduce this shortcoming by cross-checking two different registries, and thus identified additional vaccinations performed outside GP offices. Moreover, as our goal was to compare usage patterns of different SIV types, we have no reason to believe that an eventual under-registration differed by SIV type. Second, some other determinants of the use of eQIVs may have been missed, and our models may be subjected to residual confounding. For instance, we were not able to adjust for the previous season of SIV since these data were only partially available. Finally, the study was conducted in a single area during a single influenza season, and therefore its external validity/generalizability may be limited. For instance, in regions that use eQIVs to a smaller extent, these SIV types could be allocated differently. Our results should therefore be confirmed by further studies conducted in other locations and seasons.

In conclusion, in this study, we showed that a significant proportion of Italian older adults are vaccinated with standard-dose non-adjuvanted SIVs, which have been shown to be less immunogenic and effective than eQIVs. Hopefully, the recently available guidelines for the 2023/2024 season [41], in which both aQIV and hdQIV are preferentially recommended for older adults, will increase the adoption of eQIVs. The administration of different SIV types available for older adults does not occur randomly, but is dependent on individual characteristics, such as age and the presence of some co-morbidities, the period of the immunization campaign, the place of vaccination and the vaccinating physicians. These factors should be considered as potential confounders in future vaccine effectiveness research.

## Figures and Tables

**Figure 1 vaccines-11-01748-f001:**
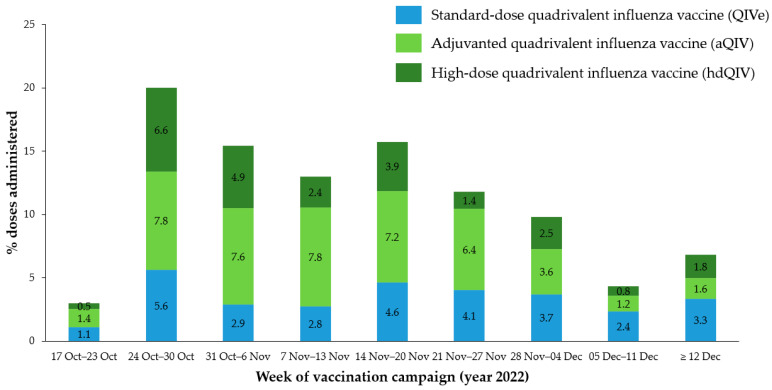
Rollout of vaccine administration, by calendar week and vaccine type (*N* = 1702).

**Table 1 vaccines-11-01748-t001:** Characteristics of the study population (*N* = 1702).

Variable	Level	% (*n*)	95% CI
Sex	Male	41.1 (700)	38.8–43.5
	Female	58.9 (1002)	56.5–61.2
Age, years	Mean (SD)	75.5	8.9 ^1^
	60–64	13.8 (235)	12.2–15.5
	65–69	17.4 (296)	15.6–19.3
	70–74	18.6 (317)	16.8–20.6
	75–79	17.9 (304)	16.1–19.8
	≥80	32.3 (550)	30.1–34.6
Chronic conditions	No	23.7 (404)	21.7–25.8
	≥1	76.3 (1298)	74.2–78.3
	Cardiovascular	64.2 (1092)	61.8–66.4
	Diabetes	19.6 (333)	17.7–21.5
	Respiratory	4.2 (72)	3.3–5.3
	Renal	7.9 (135)	6.7–9.3
	Hepatic	2.5 (42)	1.8–3.3
	Rheumatic	1.7 (29)	1.1–2.4
	Cancer and immunosuppressive	13.9 (237)	12.3–15.7
	Anemia	14.3 (244)	12.7–16.1
	Obesity	4.1 (69)	3.2–5.1
	Dementia	0.8 (13)	0.4–1.3

^1^ Standard deviation (SD); CI, confidence interval.

**Table 2 vaccines-11-01748-t002:** Comparison of the usage pattern of the enhanced and standard-dose 2022/2023 seasonal influenza vaccines among adults aged ≥ 60 years.

Variable	Level	eQIV (*N* = 1183)	sdQIV (*N* = 519)	*p*	hdQIV (*N* = 422)	aQIV (*N* = 761)	*p*
Sex	Male	40.6 (480)	42.4 (220)	0.49 ^1^	38.2 (161)	41.9 (319)	0.22 ^1^
	Female	59.4 (703)	57.6 (299)		61.8 (261)	58.1 (442)	
Age, years	Mean (SD)	77.2 (8.2)	71.5 (8.9)	<0.001 ^2^	77.4 (9.8)	77.1 (7.2)	0.59 ^2^
	60–64	5.8 (69)	32.0 (166)	<0.001 ^3^	14.7 (62)	0.9 (7) ^4^	
	65–69	15.7 (186)	21.2 (110)		13.5 (57)	17.0 (129)	
	70–74	20.1 (238)	15.2 (79)		13.0 (55)	24.0 (183)	NA ^5^
	75–79	20.4 (241)	12.1 (63)		10.4 (44)	25.9 (197)	
	≥80	38.0 (449)	19.5 (101)		48.3 (204)	32.2 (245)	
Chronic conditions	No	18.2 (215)	36.4 (189)	<0.001 ^1^	19.7 (83)	17.3 (132)	0.35 ^1^
	≥1	81.8 (968)	63.6 (330)		80.3 (339)	82.7 (629)	
	Cardiovascular	70.4 (833)	49.9 (259)	<0.001 ^1^	69.2 (292)	71.1 (541)	0.51 ^1^
	Diabetes	22.1 (262)	13.7 (71)	<0.001 ^1^	19.7 (83)	23.5 (179)	0.14 ^1^
	Respiratory	4.9 (58)	2.7 (14)	0.037 ^1^	7.1 (30)	3.7 (28)	0.011 ^1^
	Renal	9.4 (111)	4.6 (24)	<0.001 ^1^	11.4 (48)	8.3 (63)	0.095 ^1^
	Hepatic	3.0 (35)	1.3 (7)	0.061 ^1^	2.8 (12)	3.0 (23)	>0.99 ^1^
	Rheumatic	1.7 (20)	1.7 (9)	>0.99 ^1^	1.2 (5)	2.0 (15)	0.36 ^1^
	Cancer/immunosuppressive	15.0 (177)	11.6 (60)	0.068 ^1^	14.0 (59)	15.5 (118)	0.50 ^1^
	Anemia	16.7 (198)	8.9 (46)	<0.001 ^1^	18.0 (76)	16.0 (122)	0.42 ^1^
	Obesity	4.3 (51)	3.5 (18)	0.50 ^1^	4.0 (17)	4.5 (34)	0.77 ^1^
	Dementia	0.8 (10)	0.6 (3)	0.77 ^1^	1.4 (6)	0.5 (4)	0.18 ^1^
Place of vaccination	General practitioner office	92.1 (1089)	77.6 (403)	<0.001 ^1^	95.7 (404)	90.0 (685)	<0.001 ^1^
	Community pharmacy	4.0 (47)	7.9 (41)		0 (0)	6.2 (47)	
	Other	4.0 (47)	14.5 (75)		4.3 (18)	3.8 (29)	

^1^ Fisher’s exact test. ^2^
*t* test. ^3^ Chi square test. ^4^ These vaccine doses represent off-label use of aQIV. ^5^ Not applicable (NA) because of different age indications; aQIV, adjuvanted quadrivalent influenza vaccine; eQIV, enhanced quadrivalent influenza vaccine; hdQIV, high-dose quadrivalent influenza vaccine; SD, standard deviation; sdQIV, standard-dose quadrivalent influenza vaccine.

**Table 3 vaccines-11-01748-t003:** Multivariable logistic regression models ^1^ used to predict the use of enhanced 2022/2023 seasonal influenza vaccines among adults aged ≥ 60 years.

Variable	Level	aOR (95% CI) ^2^	aOR (95% CI) ^3^
Sex	Male	–	Ref
	Female	–	0.90 (0.70–1.16)
Age, years	1-year increase	1.10 (1.08–1.12)	1.10 (1.09–1.12)
Chronic conditions ^4^	Cardiovascular	1.40 (1.08–1.82)	1.36 (1.04–1.76)
	Diabetes	1.62 (1.16–2.29)	1.56 (1.12–2.21)
	Respiratory	–	1.04 (0.53–2.14)
	Renal	–	1.19 (0.70–2.09)
	Hepatic	–	1.41 (0.61–3.72)
	Rheumatic	–	0.84 (0.32–2.34)
	Cancer and immunosuppressive	–	1.28 (0.88–1.87)
	Anemia	–	1.01 (0.67–1.52)
	Obesity	–	1.32 (0.70–2.56)
	Dementia	–	0.92 (0.21–5.12)
Place of vaccination	General practitioner office	Ref	Ref
	Community pharmacy	0.40 (0.24–0.67)	0.40 (0.24–0.66)
	Other	0.16 (0.10–0.25)	0.16 (0.10–0.26)
Nagelkerke’s pseudo-*R*^2^		0.371	0.374

^1^ Models are also adjusted for general practitioner and week of vaccination campaign. ^2^ Selection of variables in the multivariable model based on minimization of the Akaike information criterion. ^3^ Fully adjusted model. ^4^ The reference category is set to the absence of a given condition. aOR, adjusted odds ratio; CI, confidence interval.

## Data Availability

Raw data may be available from the corresponding author upon reasonable request.

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
