# Peer review of "Rollout of the 2022/2023 Seasonal Influenza Vaccination and Correlates of the Use of Enhanced Vaccines among Italian Adults"

_vaccines, 2023, doi:10.3390/vaccines11121748_

Round 1
Reviewer 1 Report
Comments and Suggestions for Authors
The manuscript was written well. The data presentation is clear. It needs a fresh eye proofreading for a few minor errors, e.g. line 165, "13decreased".
The data are on the relatively simple side but related to the study question. I was wondering if cost can be a factor impact people's choices. In the US, the cost of the standard and HD vaccines is different. Please address the potential impact of cost.
Overall, it is a well-designed study with a clear presentation.
Comments on the Quality of English LanguageEnglish is fine. Proofreading is necessary.
Author Response
Comment: The manuscript was written well. The data presentation is clear. It needs a fresh eye proofreading for a few minor errors, e.g. line 165, "13decreased".
Reply: Thank you for interest in our paper. We have now proofread the whole manuscript and all spelling errors have been corrected.
Comment: The data are on the relatively simple side but related to the study question. I was wondering if cost can be a factor impact people's choices. In the US, the cost of the standard and HD vaccines is different. Please address the potential impact of cost.
Reply: Contrary to the US, influenza vaccination in Italy is free-of-charge and all vaccine types are fully reimbursed. Availability of single vaccines depends therefore on regional procurement policies and allotments. We have now clarified this fact early in the text.
Comment: Overall, it is a well-designed study with a clear presentation.
Reply: Thank you again for interest in our paper. All your comments have been addressed.
Reviewer 2 Report
Comments and Suggestions for Authors
An interesting paper about influenza vaccination in Italy, some minor changes are needed.
This is an interesting paper about influenza vaccination in Italy, only some minor changes should be included.
1) In the abstract, details should be provided about the study's population and how the patients were extracted. This information is found in the material and methods in which it is said that they come from wards or doctors.
2) On line 44, when it says currently, the year 20 23 should be indicated so that readers can temporally locate the article.
3) In the material and methods on lines 104 and 105, the codes that the variables have in the database are included. This information is unnecessary, it is not of interest that the variable is that the code one is a woman or a man, the same happens with the presence variable in which it is indicated that one is present. These codes should be ignored.
4) In table 1 it is not very clear what does it mean 75.5 8.99 could be the day or standard deviations but on the other hand in the columns it seems that the confidence interval is referenced. The design of the table should be improved to make it more understandable.
5) In table 3, the asterisks with the information on the Odds ratio adjusted with its 95% confidence interval are unnecessary; the probability does not contribute anything and can be omitted.
6) The text of the article seems to imply that vaccines can be administered in pharmacies. It would be necessary to develop this aspect in the introduction and the discussion. Since readers may be in countries where vaccinations are not carried out in pharmacies. It should be mentioned in which countries vaccinations can be carried out in pharmacies besides Italy. It would be necessary to discuss the inclusion of topics related to vaccination in the training of pharmacy degree students.
Author Response
Comment: An interesting paper about influenza vaccination in Italy, some minor changes are needed. This is an interesting paper about influenza vaccination in Italy, only some minor changes should be included.
Reply: Thank you for your interest in our paper. All your comments have been addressed.
Comment: 1) In the abstract, details should be provided about the study's population and how the patients were extracted. This information is found in the material and methods in which it is said that they come from wards or doctors.
Reply: As required, the main characteristics of the study population has been provided in the abstract. Please note, however, the maximum abstract length allowed is 200 words and we therefore had to perform some cuts.
Comment: 2) On line 44, when it says currently, the year 20 23 should be indicated so that readers can temporally locate the article.
Reply: This has been now clarified.
Comment: 3) In the material and methods on lines 104 and 105, the codes that the variables have in the database are included. This information is unnecessary, it is not of interest that the variable is that the code one is a woman or a man, the same happens with the presence variable in which it is indicated that one is present. These codes should be ignored.
Reply: As required, the coding scheme has been removed.
Comment: 4) In table 1 it is not very clear what does it mean 75.5 8.99 could be the day or standard deviations but on the other hand in the columns it seems that the confidence interval is referenced. The design of the table should be improved to make it more understandable.
Reply: We have now added a clarification on the standard deviation.
Comment: 5) In table 3, the asterisks with the information on the Odds ratio adjusted with its 95% confidence interval are unnecessary; the probability does not contribute anything and can be omitted.
Reply: As required, asterisks have been removed
Comment: 6) The text of the article seems to imply that vaccines can be administered in pharmacies. It would be necessary to develop this aspect in the introduction and the discussion. Since readers may be in countries where vaccinations are not carried out in pharmacies. It should be mentioned in which countries vaccinations can be carried out in pharmacies besides Italy. It would be necessary to discuss the inclusion of topics related to vaccination in the training of pharmacy degree students.
Reply: As suggested, we have now indicated early in the text that some (limited amount) of vaccinations are performed at pharmacies. We have also discussed the important role of pharmacists in increasing vaccination uptake and necessity of their continuous training on SIV-related issues.
Reviewer 3 Report
Comments and Suggestions for Authors
While this cross-sectional study provides valuable insights into the frequency and determinants of enhanced seasonal influenza vaccine (SIV) use among older adults in Italy for the 2022/23 season, certain limitations should be acknowledged. The absence of specific guidelines for enhanced SIV use during this period raises questions about the basis for the observed trends. Furthermore, while the study emphasizes the suboptimal adoption of enhanced SIVs in Italy, it does not delve into the reasons behind this suboptimal utilization, hindering a comprehensive understanding of the barriers to their uptake.
The introduction provides a comprehensive overview of the background, context, and objectives of the study on the usage and correlates of enhanced seasonal influenza vaccines (eQIVs) in older adults in Italy for the 2022/23 season. However, it lacks a detailed exploration of the specific factors contributing to this low coverage, which could provide crucial insights for the study. Additionally, the introduction highlights the availability of various types of seasonal influenza vaccines but falls short in discussing the reasons behind the absence of specific guidelines for their use in the 2022/23 season. Addressing these gaps would enhance the foundation for the study by providing a more nuanced understanding of the context and factors influencing influenza vaccination practices in Italy, ultimately contributing to the interpretation of the study's findings.
The methodology section of the manuscript provides a clear outline of the study design, population, variables, and statistical analysis. However, firstly, the rationale for choosing the Metropolitan City of Genoa as the study location is not explicitly justified, and potential regional variations in influenza vaccination practices are not discussed. Additionally, while the study aims to assess the determinants of enhanced seasonal influenza vaccine (eQIV) use, the criteria for selecting participants and the potential impact of these criteria on the generalizability of the findings are not thoroughly examined.
The results section of the manuscript presents a thorough analysis of the data, revealing important patterns in the utilization of enhanced seasonal influenza vaccines (eQIVs) among older adults in Italy. However, while the study reports the estimated influenza vaccine uptake in older adults (54.6%), providing a valuable metric; however, a more detailed exploration of factors influencing the overall vaccination coverage, beyond the specific focus on eQIVs, would enhance the context for interpreting the findings. The results section also lacks a discussion of potential biases introduced by the reliance on electronic health records and the cross-checking with a local health unit electronic vaccination registry.
The discussion section of the manuscript provides a comprehensive interpretation of the study's findings, emphasizing the suboptimal uptake of enhanced influenza vaccines (eQIVs) among older adults in Italy during the 2022/23 season. The discussion highlights the significance of the study for policymakers, particularly in a context where no national preferential recommendation for specific SIV types was in place. However, the section could benefit from a more nuanced exploration of the implications of the study's limitations. While the authors acknowledge the potential bias introduced by relying on electronic health records, a more detailed discussion of the impact of this limitation on the study's internal validity and generalizability would enhance the transparency of the research.
Author Response
Comment: While this cross-sectional study provides valuable insights into the frequency and determinants of enhanced seasonal influenza vaccine (SIV) use among older adults in Italy for the 2022/23 season, certain limitations should be acknowledged. The absence of specific guidelines for enhanced SIV use during this period raises questions about the basis for the observed trends. Furthermore, while the study emphasizes the suboptimal adoption of enhanced SIVs in Italy, it does not delve into the reasons behind this suboptimal utilization, hindering a comprehensive understanding of the barriers to their uptake.
Reply: As suggested, we have now fully revised the subsection of study limitations. In particular, it has been deepened and contextualized. Reasons behind the lack of Italian recommendations on the specific use of different influenza vaccine types are supposedly different. This was, indeed, the primary study question (i.e., what drives vaccination physicians in selecting a particular vaccine type on the condition of no guidelines). The most plausible reason is that National decision makers pointed to increase influenza coverage rates with any available vaccine type (probably at lower cost). On the other hand, later (for the 2023/24 season) there was a shift in National recommendations. It is likely that this shift was driven by the recently published US recommendations on the preferential use of enhanced vaccines, which showed better protection in a number of US-based studies. We have now discussed these observations and put them into international context.
Comment: The introduction provides a comprehensive overview of the background, context, and objectives of the study on the usage and correlates of enhanced seasonal influenza vaccines (eQIVs) in older adults in Italy for the 2022/23 season. However, it lacks a detailed exploration of the specific factors contributing to this low coverage, which could provide crucial insights for the study. Additionally, the introduction highlights the availability of various types of seasonal influenza vaccines but falls short in discussing the reasons behind the absence of specific guidelines for their use in the 2022/23 season. Addressing these gaps would enhance the foundation for the study by providing a more nuanced understanding of the context and factors influencing influenza vaccination practices in Italy, ultimately contributing to the interpretation of the study's findings.
Reply: As in other European countries, low influenza vaccination coverage in Italy has been mostly linked to lack of confidence in the vaccine, low perceived need for vaccination, low perceived risk of influenza, missing recommendation from health providers and inadequate social marketing campaigns. From the point of view of subject characteristics, age and presence of co-morbidities have the largest effect on influenza vaccination uptake in Italian adults. We have now described these reasons early in the manuscript. Moreover, as also suggested in another comment, we have now reported post-hoc results on the correlates of influenza vaccination (i.e., any influenza vaccine type) in our sample. These results were fully in line with the previous research. We have also provided reasons for the lack of Italian recommendations on the specific use of different influenza vaccine types (see also the reply above).
Comment: The methodology section of the manuscript provides a clear outline of the study design, population, variables, and statistical analysis. However, firstly, the rationale for choosing the Metropolitan City of Genoa as the study location is not explicitly justified, and potential regional variations in influenza vaccination practices are not discussed. Additionally, while the study aims to assess the determinants of enhanced seasonal influenza vaccine (eQIV) use, the criteria for selecting participants and the potential impact of these criteria on the generalizability of the findings are not thoroughly examined.
Reply: Thank you for this comment. Among other Italian (as well as European) cities, Genoa had the highest incidence of the elderly. Indeed, 29% of the total population is represented by adults aged ≥ 65 years. The Metropolitan City of Genoa (Liguria region in general) is among few Italian regions, in which all influenza vaccine types are used to significant extent. In other regions, for instance, procurement of the novel high-dose vaccine is limited to a very small number of doses. The selected area of Genoa therefore represents an optimal case study for policy makers. In Italy (as well as in Genoa), general practitioners (GPs) are free to order any available vaccine type (and their associated quantity) and independently of the availability of guidelines. To participate in the study, GPs had to use all vaccine types. The text has been amended accordingly. As we clearly stated among the study limitations, our results may be not generalizable to other spatial or temporal realities. We have now further enhanced this study shortcoming.
Comment: The results section of the manuscript presents a thorough analysis of the data, revealing important patterns in the utilization of enhanced seasonal influenza vaccines (eQIVs) among older adults in Italy. However, while the study reports the estimated influenza vaccine uptake in older adults (54.6%), providing a valuable metric; however, a more detailed exploration of factors influencing the overall vaccination coverage, beyond the specific focus on eQIVs, would enhance the context for interpreting the findings. The results section also lacks a discussion of potential biases introduced by the reliance on electronic health records and the cross-checking with a local health unit electronic vaccination registry.
Reply: As we mentioned earlier, the entire subsection of the study limitations has been revised. The local health unit electronic vaccination registry used in the study has been validated in two previous studies. The text has been amended accordingly. As suggested, we have now performed a separate post-hoc analysis on the determinants of vaccination uptake in older adults (see §1 of the Results and Supplementary Table S1). As expected, vaccination uptake was associated with higher age and presence of several comorbidities. Please note, however, that the study was not designed nor powered to establish correlates of any vaccine uptake.
Comment: The discussion section of the manuscript provides a comprehensive interpretation of the study's findings, emphasizing the suboptimal uptake of enhanced influenza vaccines (eQIVs) among older adults in Italy during the 2022/23 season. The discussion highlights the significance of the study for policymakers, particularly in a context where no national preferential recommendation for specific SIV types was in place. However, the section could benefit from a more nuanced exploration of the implications of the study's limitations. While the authors acknowledge the potential bias introduced by relying on electronic health records, a more detailed discussion of the impact of this limitation on the study's internal validity and generalizability would enhance the transparency of the research.
Reply: As suggested, we have now provided (early in the Discussion) more context on the suboptimal uptake of enhanced vaccines. Study limitations have been further described and exemplified.
Reviewer 4 Report
Comments and Suggestions for Authors
This paper reports a descriptive statistical analysis of the 2022/23 seasonal influenza vaccination and correlates of the use of enhanced vaccines among Italian adults. Overall, the paper is well written and the analyses follow standard statistical protocols for such analyses. Here are a couple of items to attend to in a revision.
First, the first and third sentences of the paragraph in the Results section that begins in line 137 refer to sample numbers statistics that are combined from rows of Table 2. It would be good to include rows in the table that report these as well.
Second, Table 2 reports statistical significance values in the column on the right side that are located between rows slightly above and slightly below those values. It would be good to indicate the statistical comparisons/tests to which these significance values are referenced.
One of the main takeaways for me from your research findings is the importance of the GPs and their recommendations (clear articulations and consistencies of their messages) to the members of your sample.
Author Response
Comment: This paper reports a descriptive statistical analysis of the 2022/23 seasonal influenza vaccination and correlates of the use of enhanced vaccines among Italian adults. Overall, the paper is well written and the analyses follow standard statistical protocols for such analyses. Here are a couple of items to attend to in a revision.
Reply: Thank you for your interest in our paper. All your comments have been addressed.
Comment: First, the first and third sentences of the paragraph in the Results section that begins in line 137 refer to sample numbers statistics that are combined from rows of Table 2. It would be good to include rows in the table that report these as well.
Reply: As suggested, these numbers have been added to Table 2.
Comment: Second, Table 2 reports statistical significance values in the column on the right side that are located between rows slightly above and slightly below those values. It would be good to indicate the statistical comparisons/tests to which these significance values are referenced.
Reply: Thank you for this observation. We have now added horizontal lines to distinguish between tests. We have also specified on which test has been used for each comparison.
Comment: One of the main takeaways for me from your research findings is the importance of the GPs and their recommendations (clear articulations and consistencies of their messages) to the members of your sample.
Reply: We agree with your comment. Indeed, the important role of GPs has been stressed.
Round 2
Reviewer 3 Report
Comments and Suggestions for Authors
The authors have addressed the suggestion and have significantly improved the manuscript. It is my pleasure to recommend the manuscript for publication in it present form.